# Direct Interactions with Nascent Transcripts Is Potentially a Common Targeting Mechanism of Long Non-Coding RNAs

**DOI:** 10.3390/genes11121483

**Published:** 2020-12-10

**Authors:** Ivan Antonov, Yulia Medvedeva

**Affiliations:** 1Research Center of Biotechnology, Institute of Bioengineering, Russian Academy of Science, 119071 Moscow, Russia; ivan.antonov@gatech.edu; 2Department of Biological and Medical Physics, Moscow Institute of Physics and Technology, Dolgoprudny, 141701 Moscow Region, Russia

**Keywords:** long non-coding RNAs, RNA-RNA interactions, nascent transcripts, gene regulation

## Abstract

Although thousands of mammalian long non-coding RNAs (lncRNAs) have been reported in the last decade, their functional annotation remains limited. A wet-lab approach to detect functions of a novel lncRNA usually includes its knockdown followed by RNA sequencing and identification of the deferentially expressed genes. However, identification of the molecular mechanism(s) used by the lncRNA to regulate its targets frequently becomes a challenge. Previously, we developed the ASSA algorithm that detects statistically significant inter-molecular RNA-RNA interactions. Here we designed a workflow that uses ASSA predictions to estimate the ability of an lncRNA to function via direct base pairing with the target transcripts (co- or post-transcriptionally). The workflow was applied to 300+ lncRNA knockdown experiments from the FANTOM6 pilot project producing statistically significant predictions for 71 unique lncRNAs (104 knockdowns). Surprisingly, the majority of these lncRNAs were likely to function co-transcriptionally, i.e., hybridize with the nascent transcripts of the target genes. Moreover, a number of the obtained predictions were supported by independent iMARGI experimental data on co-localization of lncRNA and chromatin. We detected an evolutionarily conserved lncRNA *CHASERR* (AC013394.2 or LINC01578) that could regulate target genes co-transcriptionally via interaction with a nascent transcript by directing CHD2 helicase. The obtained results suggested that this nuclear lncRNA may be able to activate expression of the target genes in trans by base-pairing with the nascent transcripts and directing the CHD2 helicase to the regulated promoters leading to open the chromatin and active transcription. Our study highlights the possible importance of base-pairing between nuclear lncRNAs and nascent transcripts for the regulation of gene expression.

## 1. Introduction

Long non-coding RNAs (lncRNAs)—transcripts of at least 200 nucleotides long lacking protein-coding potential—are the most populated class of transcripts in the mammalian genomes. They are expressed from more than 50,000 loci in the human genome [1], which is almost twice the number of protein-coding loci. LncRNAs are usually less conserved [2], lower expressed, and more cell-type-specific [1] compared to protein-coding genes. Still, the transcription of lncRNAs is regulated suggesting their functionality [1,3]. Several hundred human lncRNAs have been extensively characterized so far [4,5,6], revealing their roles in regulating transcription [7] and chromatin state [8,9,10,11]. However, the majority of the non-coding transcriptome remains non-annotated.

Up to date, several genomic screening approaches to functional annotation of lncRNA have been developed. Based on lncRNA knockdown (knockout) followed by cellular or molecular phenotyping, these approaches characterize a fair share of lncRNAs in one series of experiments [12,13,14,15]. In the FANTOM6 pilot project [12] more than 300 RNAs expressed in human primary dermal fibroblasts (HDFs) were suppressed using antisense LNA-modified GapmeR antisense oligonucleotide (ASO) technology. The effect of each knockdown was estimated by Cap Analysis Gene Expression (CAGE) deep sequencing to reveal molecular pathways associated with each lncRNA.

Complementary approaches to lncRNA annotation aim to search for their targets via detecting interactions between lncRNAs and chromatin or other RNA molecules. LncRNAs use different molecular mechanisms to bind to the chromatin, including RNA-DNA binding mediated by protein complexes, direct RNA-DNA hybridization via triplexes, RNA binding to single-stranded DNA regions (known as R-loops) and co-transcriptional RNA-RNA interactions via duplexes with nascent RNA transcripts (reviewed in [16]). Experimental techniques to detect RNA:chromatin interactions (ChIRP-seq [17], MARGI [18], ChAR-seq [19], GRID-seq [20], RADICL-seq [21], Red-C [22]) usually cannot differentiate between various mechanisms of RNA:chromatin interactions and focus solely on the location of such interaction. Direct RNA:RNA interactions can be detected with SPLASH [23], PARIS [24], LIGR-seq [25], MARIO [26] and other methods. Yet, the majority of the contacts obtained by such methods reflect RNA structure leading to very limited information on the inter-molecular lncRNA:RNA interactions available. Due to these limitations computational approaches for detecting RNA:RNA interactions provide valuable information for lncRNA functional annotation.

In this work, we focus on RNA:RNA interactions. We investigated two possible scenarios— lncRNA bound co-transcriptionally to the nascent RNAs via duplexes and lncRNA bound to mature transcripts—by applying the ASSA RNA-RNA interaction prediction tool [27] to the data on expression changes after lncRNA knockdown available from the FANTOM6 pilot project. Therefore, the present study pursued two goals: (i) to identify lncRNAs that are likely to function via direct RNA:RNA interactions with its targets and (ii) for the identified lncRNAs to predict the most likely interaction mode (co-transctiptional or post-transcriptional).

## 2. Materials and Methods

### 2.1. Computational Prediction of Inter-Molecular RNA:RNA Interactions

The AntiSense Search Approach (ASSA, https://github.com/vanya-antonov/assa) tool was used in this study to predict inter-molecular RNA:RNA interactions [27]. We decided to use ASSA because previously its performance has been evaluated on several well studied lncRNA data sets (RIA-seq for TINCR lncRNA [28], SPLASH for 50 different RNAs [23] as well as 17 lncRNA:RNA interactions from selected publications). We compared the results obtained by ASSA with 17 different RNA:RNA prediction tools. Although accurate computational prediction remains tricky, ASSA outperformed all other tested tools (see [16] for more details).

### 2.2. Distinguishing Co-Transcriptional and Post-Transcriptional Interactions

To distinguish co-transcriptional and post-transcriptional modes of lncRNA action we prepared two sets of target RNA sequences. We assumed that in the co-transcriptional mode the lncRNA can directly base pair with the nascent transcripts. Thus, the co-transcriptional lncRNA-RNA interactions may regulate gene expression at the transcriptional level. Since the regulation of transcription frequently occurs via gene promoters, we considered the 5’ end of the nascent transcript only. Specifically, we predicted possible interactions between query mature lncRNA (without introns) with the first 1kb of the target gene (including the promoter and all the introns, if any). We included the introns assuming that lncRNA-RNA binding may precede splicing.

By contrast, post-transcriptional lncRNA:RNA hybridizations occur with mature spliced target transcripts in the cytoplasm or the nucleus (depending on cellular localization of the lncRNA). We hypothesized that this type of interaction results in post-transcriptional regulation of gene expression. It should be noted that CAGE-seq technology which has been used in the FANTOM6 project measures expression changes at the promoter level. Thus, to predict post-transcriptional interactions we considered the longest isoform that can be expressed from the corresponding promoter only.

In the present work ASSA was run with default settings in both the co- and post-transcriptional modes.

### 2.3. Statistical Analysis

To estimate the correspondence between the iMARGI peaks and ASSA predictions for each lncRNA we only considered differentially expressed genes (DEGs) that had predicted antisense interactions with the corresponding lncRNA (ASSA *p*-values <0.01). It should be noted that the fraction of the ASSA hits among all the DEGs for a particular lncRNA varied depending on the knockdown experiments.

For each lncRNA correspondence between the DEGs with the predicted ASSA hits and the experimentally identified lncRNA:DNA interactions (iMARGI peaks [18]) were computed using the GenometriCorr R package [29]. The tool computes the statistical significance of the observed co-localization between two sets of genomic intervals. The *p*-values were calculated based on the 100 permutations.

The statistical significance of the gene overlaps was estimated using the hypergeometric distribution (the phyper R function). We used 45480 as the total number of the human genes that was computed as the sum of the protein-coding (19954), long (17957) and small (7569) non-coding RNA genes as annotated in the GENCODE version 35. The human orthologs of the DEGs Chaserr-/- mEFs [30] were identified using the getLDS function of the biomaRt R package. The expression of the CHD2 gene was obtained from the FANTOM5 SSTAR online tool (https://fantom.gsc.riken.jp/5/sstar/EntrezGene:1106) for the “Fibroblast-skin normal, donor2 (nuclear fraction).CNhs12582.14302-155B9” sample.

The code and the corresponding data files that were used to generate the images in the manuscript are located in Appendix A and also available at https://github.com/vanya-antonov/article_assa_and_f6.

## 3. Results

### 3.1. lncRNA Knockdown Experiments Suggest a Widespread Regulation Via RNA:RNA Interactions

We used the AntiSense Search Approach (ASSA) tool [27] to identify lncRNAs that were likely to regulate gene expression via direct hybridization with the target transcripts. In our previous benchmarking study, we have shown that ASSA outperformed other computational tools both in accuracy and in execution time [16]. In addition to computing the free energy of inter-molecular interaction between two RNAs, ASSA estimates its statistical significance (adjusted *p*-values) taking into account the lengths and the GC contents of both transcripts.

We analyzed all the lncRNAs knockdown experiments available from the recent pilot FANTOM6 project [12]. In brief, for each lncRNA query we investigated the enrichment of strong ASSA hits among the differentially expressed genes (DEGs) after the knockdown of this lncRNA. We considered two possible scenarios of lncRNA antisense interactions—co- and post-transcriptional modes. To do so we prepared two sets of the target RNA sequences: (i) either the 1 kb downstream regions (including introns, if any) starting from the target promoters (co-transcriptional mode) or (ii) the mature (i.e., without introns) RNA transcripts produced from the corresponding promoters (post-transcriptional mode). This allowed us to investigate possible lncRNA functioning in different cellular compartments–nucleus and cytoplasm (see Figure 1 and Methods for more details).

Among the 337 analyzed knockdown experiments statistically significant enrichment in the number of ASSA hits in the DEGs was identified for 98 and 47 ASOs in the co- and post-transcriptional mode, respectively, suggesting possible regulation of these DEGs via lncRNA:RNA interactions (see Figure 2A,B). The number of the cases of co-transcriptional interactions was more than twice higher than that of the post-transcriptional interactions. Interestingly, the majority (41 out of the 47) of the ASO knockdowns that showed enrichment in the post-transcriptional mode had enrichment in the co-transcriptional mode as well (Figure 2C). These 41 knockdown experiments corresponded to 31 different lncRNAs. It seemed unlikely that all these lncRNAs were able to function in both modes at the same time. To determine the molecular mechanism that was more likely for those 31 lncRNAs we used independent experimental data on the global RNA-chromatin interactions obtained by the iMARGI technology [18,31]. We investigated the co-localization between the iMARGI peaks for *CHASERR* and the promoters that were differentially expressed upon its knockdown and detected strong ASSA hits in the co-transcriptional mode [29]. Statistically significant co-localization (*p*-value < 0.01) was observed for 21 of those ambiguous 41 ASO knockdowns (Figure 2D) indicating that the corresponding lncRNAs were likely to function co-transcriptionally. Altogether, 35% and 14% of the FANTOM6 knockdowns showed enrichment of the ASSA hits among the DEGs in the co- and post-transcriptional modes, respectively. This indicated that direct hybridization with the target transcripts may be a common molecular mechanism of the lncRNA action.

### 3.2. Evolutionary Conserved *CHASERR* (AC013394.2) lncRNA May Direct the CHD2 Helicase to the Specific Genomic Loci by Interacting with Nascent Transcripts

Two knockdown experiments for the AC013394.2 lncRNA (also known as CHASERR, LINC01578 or LOC100507217) demonstrate coordinated and strong enrichment of the ASSA predicted RNA:RNA duplexes in DEG (Figure 2D). The predicted RNA:RNA contacts are supported by iMARGI co-localization data (Appendix A). It should be noted that in the FANTOM6 project there were three independent ASO knockdowns of the CHASERR lncRNA. However, only two of them (ASO_G0272888_AD_07 and ASO_G0272888_AD_10) can be considered successful as the third experiment (ASO_G0272888_04) did not result in a significant reduction of CHASERR levels (Appendix A). Importantly, the two successful knockdowns resulted in a change of expression for similar sets of genes. Specifically, more than 50% of the DEGs identified in the ASO_G0272888_AD_10 experiment were also detected in the ASO_G0272888_AD_07 (Appendix A, hypergeometric test *p*-value = 0). This indicated the *CHASERR* functional significance and reproducibility of the results.

It has recently been shown that the mouse ortholog of *CHASERR* is mainly localized in the nucleus where it is bound to the chromodomain helicase DNA binding protein 2 (Chd2) [30]. The chromatin-remodeling enzyme CHD2 facilitates disassembly, eviction, sliding, and spacing of nucleosomes [32]. Thus, CHD2 localization to gene promoters opens the chromatin, increases DNA accessibility and expression [33]. Importantly, the primary sequences of human and mouse *CHASERR* orthologs are highly similar to each other (Appendix A). To check the conservation of the CHASERR functionality we obtained human orthologs of all the DEGs identified in the *Chaserr*-/- mouse embryonic fibroblases (mEFs) (see Methods) and compared them to the lists of the DEGs from the two FANTOM6 knockdowns. We observed a statistically significant overlap between the three sets of genes (hypergeometric test *p*-values = 0) supporting conservation of the CHASERR target genes (Figure 3A).

Taking into account the sequence and functional similarities between human and mouse *CHASERR* orthologs as well as the fact that the CHD2 helicase is also expressed in the human fibroblasts (FANTOM5 expression = 113.25), we assumed that *CHASERR* lncRNA and CHD2 helicase interact in the nucleus of human fibroblasts as well. Our computational analysis suggested that *CHASERR* may directly interact with the beginning (the 5${}^{\prime }$ part) of the target nascent transcripts and thus localize the CHD2 protein to the promoters of the corresponding genes. The subsequent opening of the chromatin would allow better access for the transcription factors to the promoter regions and activation of the gene expression. Indeed, both *CHASERR* knockdown experiments in the human fibroblasts resulted mainly in down-regulation of the target genes (Appendix A). Moreover, 126 (57.5%) of the 219 DEGs that were common between the two knockdowns and the *Chaserr*-/- mEFs showed expression reduction in all the three experiments (Figure 3B). Finally, it has been experimentally shown that CHASERR interacts with transcripts of 7 different genes in Hela and HEK293T human cells [24,34]. Two of these 7 genes (TNPO2 and NUP153) were present among the 3599 differentially expressed genes identified in the two successful CHASERR knockdowns in the FANTOM6 project (hypergeometric test *p*-values = 0.0136). It should also be noted that ASSA predicted statistically significant interactions between CHASERR and transcripts of these two genes (with *p*-values 2.45 × 10${}^{-8}$ and 1.66 × 10${}^{-12}$ respectively). Altogether, these results indicated that *CHASERR* mainly enhanced the expression of the target genes possibly by interacting with the nascent transcripts and directing the CHD2 helicase to corresponding promoter regions.

## 4. Discussion and Conclusions

The main goal of the recent FANTOM6 pilot project was to perform a functional annotation of hundreds of human long non-coding RNAs expressed in human fibroblasts [12]. Although a tremendous amount of work has been done in performing thousands of lncRNA knockdown experiments, subsequent interpretation of the obtained results turned out to be a challenging task. There were several important aspects that complicated the bioinformatics analysis, including (i) relatively low expression of the majority of the selected lncRNAs, (ii) inability of some of the ASOs to efficiently reduce the levels of the corresponding lncRNAs, (iii) non-specific cell response to the performed perturbations, (iv) wide variety of different molecular mechanisms that can be used by lncRNAs to regulate target genes, (v) lack of computationally efficient bioinformatics tools that can produce reliable predictions for lncRNAs.

In the present study we investigated possible regulation via direct lncRNA:RNA interactions. To do so we used ASSA [27]–a tool that not only allows analysis of thousands of possible interactions in a reasonable execution time but also provides an unbiased estimate of the statistical significance of each prediction [16]. The design of the current analysis allowed us to distinguish putative co-transcriptional and post-transcriptional lncRNA interactions. Around 30% of all the FANTOM6 knockdown experiments showed a statistically significant enrichment of lncRNA:RNA duplexes among the DEGs indicating that corresponding lncRNAs may regulate their target genes via direct base-pairing with the transcripts. Surprisingly, the majority of such lncRNAs were likely to function in the nucleus where they can interact with the nascent RNAs co-transcriptionally. This observation was also supported with the co-localization of the predicted lncRNA-RNA interactions and the lncRNA-chromatin peaks experimentally identified by iMARGI technology.

Strong enrichment of predicted lncRNA:RNA duplexes in the DEGs were observed in both successful knockdowns of the *CHASERR* lncRNA. Moreover, ASSA predictions for these two experiments correlated well with the iMARGI peaks suggesting that *CHASERR* may regulate its target genes via RNA:RNA interactions co-transcriptionally. Finally, it has previously been shown experimentally that CHD2 had distinct binding profiles across active TSS regions [35]. The authors have concluded that transcription-coupled recruitment of CHD2 occurs at transcribed gene TSSs. Our bioinformatics analysis led us to the same conclusion and highlighted the role of *CHASERR* lncRNA in this process.

In an attempt to additionally verify the co-transcriptional RNA:RNA interactions we analyzed 9 experimental interactions of CHASERR detected by PARIS method. Unfortunately, these interactions were observed in Hela and HEK293T cells, while FANTOM6 pilot data was obtained in fibroblasts. To the best of our knowledge cell-type specificity of RNA:RNA contacts has never been thoroughly investigated. Yet, lncRNA expression is highly tissue specific, suggesting that contacts obtained in one cell type may not be preserved in another one. On top of that, not all the contacts may lead to functional consequences such as change of gene expression. Given that, the presence of experimental support for two genes TNPO2 and NUP153 is remarkable.

Importantly, a recent study has shown that the *Chaserr* mouse ortholog is indeed localized in the nucleus and regulates gene expression via interactions with chromatin [30] supporting the validity of our approach. Moreover, in the original study the authors mainly focused on the cis-action of *Chaserr*. Our results suggested that *CHASERR* may be able to enhance gene expression in trans by directing the CHD2 helicase to specific genomic loci. The model of CHASERR-guided gene regulation is provided in Figure 4. Since *CHASERR* enhances gene expression, its knockdown would lead to decreased expression of the regulated genes - the result we obtained by analyzing FANTOM6 data.

To conclude, our global analysis of the FANTOM6 experimental data showed that lncRNA:RNA interactions may be a common targeting mechanism to regulate gene expression (mainly co-transcriptionally). A more detailed analysis of the *CHASERR* lncRNA suggests the role of lncRNA:nascent RNA co-transcriptional interactions in gene regulation in trans. Therefore, future studies of the nuclear lncRNAs and their interactions with the chromatin may shed a new light on their regulatory mechanisms and further improve their functional annotation.

## Figures and Tables

**Figure 1 genes-11-01483-f001:**
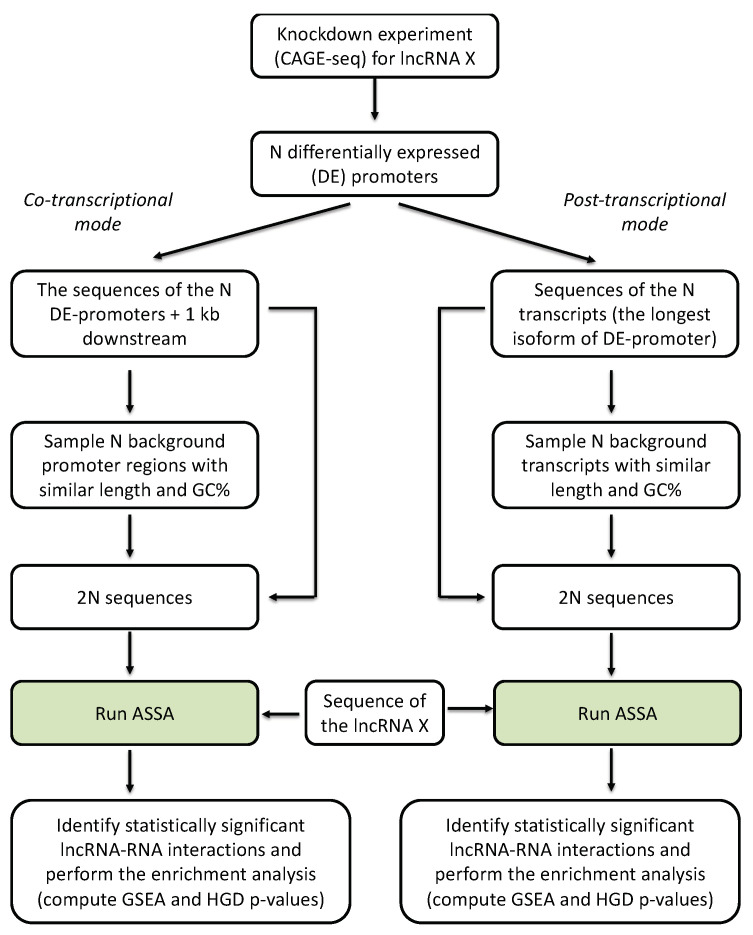
The design of the workflow to detect lncRNAs that may regulate target genes via direct base-pairing co- or post-transcriptionally.

**Figure 2 genes-11-01483-f002:**
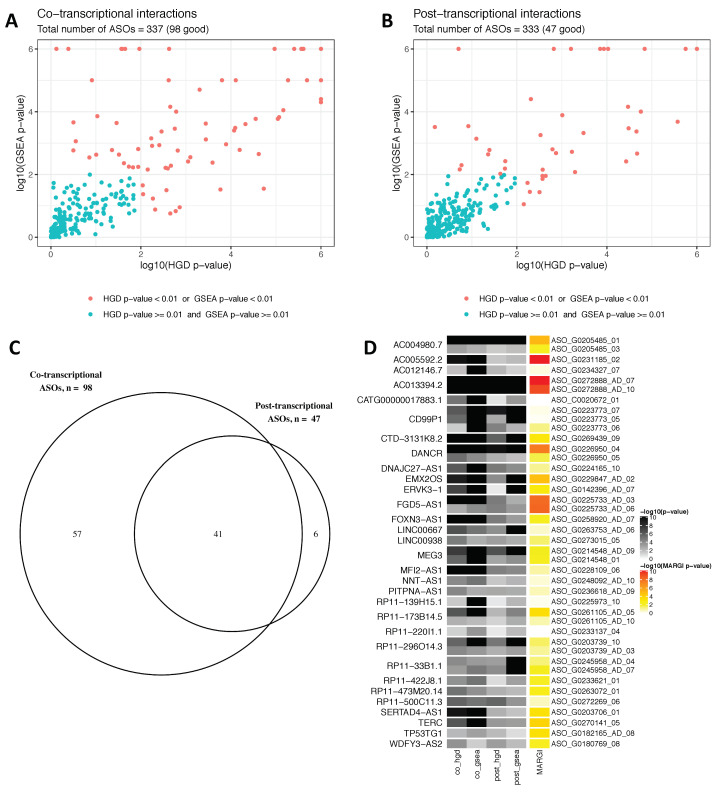
Enrichment of the lncRNA hybridization targets predicted by ASSA among differentially expressed genes identified in the FANTOM6 ASO knockdown experiments in two modes. (**A**,**B**) ASSA was applied to the differentially expressed and background transcripts assuming co-transcriptional or post-transcriptional lncRNA:RNA interaction mode. The-$\log_{10}$ of enrichment *p*-values were computed using the hypergeometric (HGD) and GSEA tests. Knockdowns with either *p*-value <0.01 are marked by the red color. There were 98 and 47 such ASOs for the co- and post-transcriptional modes, respectively. (**C**) The majority (41 out of the 47) of the ASOs with significant *p*-values in the post-transcriptional mode also had significant *p*-values in the co-transcriptional mode. (**D**) The heatmap showing four *p*-values for the 41 ASOs that produced strong *p*-values in both the co- and the post-transcriptional modes. The 41 ASO ids are on the right side and the corresponding 31 lncRNA gene names are on the left side of the heatmap. The *p*-values of the co-occurrence with the iMARGI peaks are shown on the right.

**Figure 3 genes-11-01483-f003:**
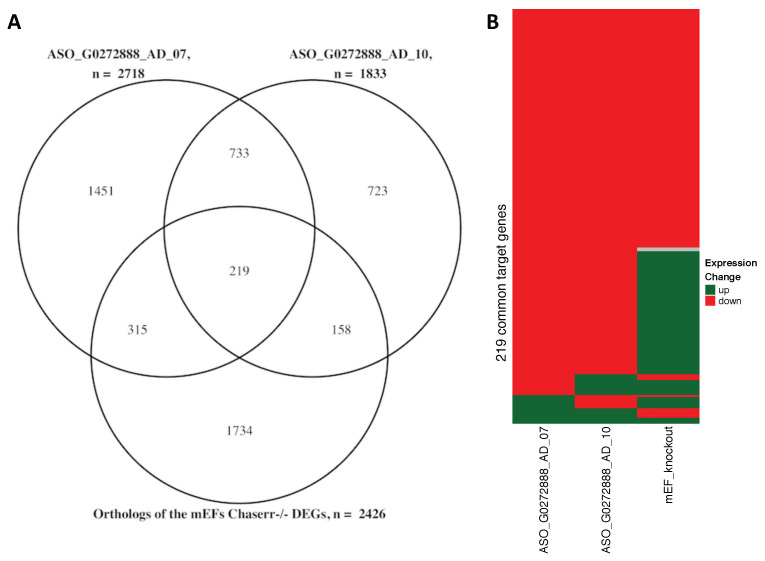
Differentially expressed genes (DEGs) observed in *CHASERR* knockdown experiments in human cells and in *Chaserr*-/- mouse embryonic fibroblasts (mEFs). (**A**) Overlap between the DEGs from three independent experiments. (**B**) Expression changes of the 219 genes that were classified as differentially expressed in all the three cases.

**Figure 4 genes-11-01483-f004:**
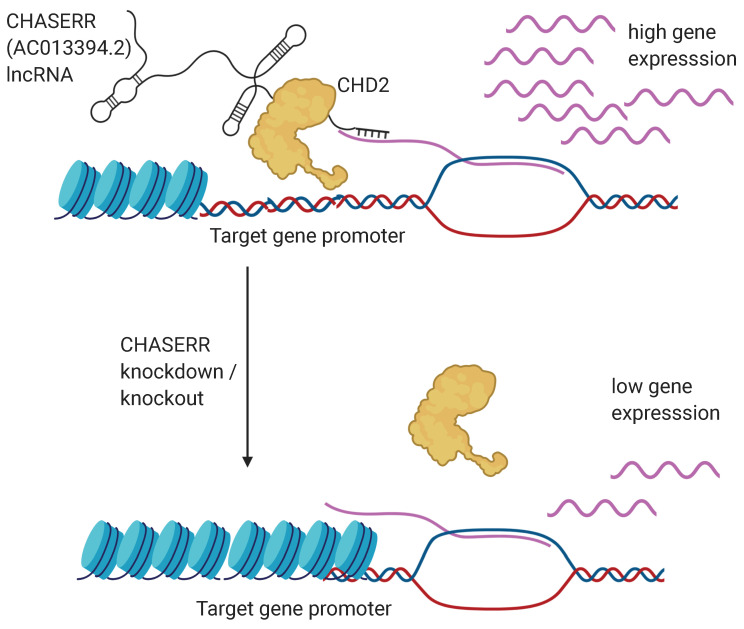
The proposed model of the CHASERR lncRNA regulation of the target genes involving the CHD2 protein and the RNA-RNA interactions with the nascent transcripts. The figure was created with BioRender.com online tool.

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
