# Peer review of "Direct Interactions with Nascent Transcripts Is Potentially a Common Targeting Mechanism of Long Non-Coding RNAs"

_genes, 2020, doi:10.3390/genes11121483_

Round 1
Reviewer 1 Report
Antonov and Medvedeva report the outcome of a bioinformatics analysis that focuses on the identification of regulatory lncRNAs, which exert their function via intermolecular RNA-RNA interactions. The authors used their own RNA-RNA interaction prediction tool (ASSA), published RNA profiling data from large-scale lncRNA knock-down experiments and lncRNA/chromatin co-localisation data to identify candidate RNAs. Among 31 promising cases they focus on CHASERR. This lncRNA acts in cis on the down-stream chromatin remodeling enzyme CHD2, presumably via an epigenetic effect caused by CHASERR’s transcription across the CHD2 promoter region. The manuscript is very well written and nicely illustrated.
The key finding of the report by Antonov and Medvedeva are RNA-RNA interactions in trans between antisense CHASERR and overlapping sense RNAs. The authors used their ASSA tool to identify these interactions and the result is certainly intriguing. However, additional data showing pervasive interactions between CHASERR and other transcripts would lend further credibility to the author’s claims that CHASERR indeed plays a more general role in the regulation of gene expression mediated by chromatin remodeling.
I suggest integrating RNA-RNA interaction data based not only on predictions but on experimentation, such as those provided by the RISE database, which references nine interactions for CHASERR, including eight mRNAs and one lncRNA (http://rise.zhanglab.net; Gong et al., Nucleic Acids Res 2017; use LINC01578 as a query). Are any of these transcripts among potential CHASERR targets reported by the authors?
Another key piece of evidence would be if CHD2 were indeed recruited to the promoters of the novel trans targets; see Siggens et al., Epigenetics Chromatin 2015 for data on transcription-coupled recruitment of CHD1 and CHD2 to transcription start sites.
Finally, I think the authors should address the regulatory mechanism they propose in somewhat more detail. Most of the target genes represented in Figure 3B are down-regulated. The model in Figure 4, however, describes CHASERRR/CHD2-mediated up-regulation. It would be helpful to include more extensive information on the known roles of CHD2 in (up/down) regulating gene expression. In line 111 the authors cite Rom et al., 2019 as a paper that shows CHASERR-mediated gene activation but the study demonstrates that CHD2 is down-regulated, potentially via promoter interference. Perhaps Semba et al., Nucleic Acids Res 2017 would be more appropriate; the authors report that the enzyme prevents the formation of suppressive chromatin.
Minor comment:
Line 57: co-transctiptional
Author Response
=== Reviewer 1 ====
Antonov and Medvedeva report the outcome of a bioinformatics analysis
that focuses on the identification of regulatory lncRNAs, which exert
their function via intermolecular RNA-RNA interactions. The authors
used their own RNA-RNA interaction prediction tool (ASSA), published
RNA profiling data from large-scale lncRNA knock-down experiments and
lncRNA/chromatin co-localisation data to identify candidate RNAs.
Among 31 promising cases they focus on CHASERR. This lncRNA acts in
cis on the down-stream chromatin remodeling enzyme CHD2, presumably
via an epigenetic effect caused by CHASERR’s transcription across the
CHD2 promoter region. The manuscript is very well written and nicely
illustrated.
The key finding of the report by Antonov and Medvedeva are RNA-RNA
interactions in trans between antisense CHASERR and overlapping sense
RNAs. The authors used their ASSA tool to identify these interactions
and the result is certainly intriguing. However, additional data
showing pervasive interactions between CHASERR and other transcripts
would lend further credibility to the author’s claims that CHASERR
indeed plays a more general role in the regulation of gene expression
mediated by chromatin remodeling.
I suggest integrating RNA-RNA interaction data based not only on
predictions but on experimentation, such as those provided by the RISE
database, which references nine interactions for CHASERR, including
eight mRNAs and one lncRNA (http://rise.zhanglab.net; Gong et al.,
Nucleic Acids Res 2017; use LINC01578 as a query). Are any of these
transcripts among potential CHASERR targets reported by the authors?
Response:
We would like to thank the reviewer for the high evaluation of our work and for pointing our attention to this valuable database. Indeed, there were 9 interactions for CHASERR that were identified experimentally by the PARIS method. These interactions corresponded to 7 unique genes (MEGF8 and NUP153 had 2 different interacting regions). Two of these 7 genes (TNPO2 and NUP153) were present among the 3599 differentially expressed genes (hypergeometric test p-value = 0.0136) identified in the two successful CHASERR knockdowns in the FANTOM6 project. Moreover, NUP153 had two different interacting sites confirmed by PARIS and was also identified as differentially expressed in one of the FANTOM6 knockdowns. It should be noted that ASSA predicts statistically significant interactions between CHASERR and both the TNPO2 and NUP153 transcripts (with p-values 2.45e-08 and 1.66e-12 respectively). Unfortunately, PARIS method was performed in Hela and HEK293T cells, while FANTOM6 pilot data was obtained in fibroblasts. To the best of our knowledge cell-type specificity of RNA:RNA contacts has never been thoroughly investigated. Yet, lncRNA expression is highly tissue specific suggesting that contacts obtained in one cell type may not be preserved in another one. On top of that, not all the contacts may lead to functional consequences such as change of gene expression. Given that, the presence of experimental support for two genes is remarkable.
We added the following text to the paper
to the results:
Finally, it has been experimentally shown that CHASERR interacts with transcripts of 7 different genes in Hela and HEK293T human cells \cite{Lu:2016aa, gong2018rise}. Two of these 7 genes (TNPO2 and NUP153) were present among the 3599 differentially expressed genes identified in the two successful CHASERR knockdowns in the FANTOM6 project (hypergeometric test p-value = 0.0136). It should also be noted that ASSA predicted statistically significant interactions between CHASERR and transcripts of these two genes (with p-values 2.45e-08 and 1.66e-12 respectively).
to the discussion:
Unfortunately, experimental interactions of CHASERR detected by PARIS method were observed in Hela and HEK293T cells, while FANTOM6 pilot data was obtained in fibroblasts. To the best of our knowledge cell-type specificity of RNA:RNA contacts has never been thoroughly investigated. Yet, lncRNA expression is highly tissue specific suggesting that contacts obtained in one cell type may not be preserved in another one. On top of that, not all the contacts may lead to functional consequences such as change of gene expression. Given that, the presence of experimental support for two genes TNPO2 and NUP153 is remarkable.
Another key piece of evidence would be if CHD2 were indeed recruited
to the promoters of the novel trans targets; see Siggens et al.,
Epigenetics Chromatin 2015 for data on transcription-coupled
recruitment of CHD1 and CHD2 to transcription start sites.
Response:
We are extremely grateful to the reviewer for mentioning this article. We believe that it indeed further supported our bioinformatics observations.
We added the following text to the Discussion:
Finally, it has previously been shown experimentally that CHD2 had distinct binding profiles across active TSS regions. The authors have concluded that transcription-coupled recruitment of CHD2 occurs at transcribed gene TSSs. Our bioinformatics analysis led us to the same conclusion and highlighted the role of \textit{CHASERR} lncRNA in this process.
Finally, I think the authors should address the regulatory mechanism
they propose in somewhat more detail. Most of the target genes
represented in Figure 3B are down-regulated. The model in Figure 4,
however, describes CHASERRR/CHD2-mediated up-regulation. It would be
helpful to include more extensive information on the known roles of
CHD2 in (up/down) regulating gene expression.
Response:
We do not see a contradiction here. Since the presence of CHASERR/CHD2 mediate gene activation, the absence (or reduced levels) of CHASERR obtained in the knockdown experiments lead to downregulation of the genes.
To avoid further confusion we added the following text to the discussion:
The model of CHASERR-guided gene regulation is provide in Figure \ref{fig:model}. Since \textit{CHASERR} enhances gene expression, its knockdown would lead to decreased expression of the regulated genes - the result we obtained by analysing FANTOM6 data.
In line 111 the authors cite Rom et al., 2019 as a paper that shows CHASERR-mediated gene
activation but the study demonstrates that CHD2 is down-regulated,
potentially via promoter interference. Perhaps Semba et al., Nucleic
Acids Res 2017 would be more appropriate; the authors report that the
enzyme prevents the formation of suppressive chromatin.
Response:
We would like to thank the reviewer for pointing us to the more relevant paper -- we have updated the reference in the manuscript.
Reviewer 2 Report
The authors in this work use their previously developed tool ASSA to identify direct RNA-RNA interactions of lncRNAs. While development of such tool to make accurate predictions is of great interest for the lncRNA biology field, the paper in the current form under delivers what is claimed in the abstract.
a) The writing is not lucid particularly the results and method section.
b) ASSA should be applied to well-studied lncRNAs like Xist, rox for which CHIRP datasets are available to validate the targets predicted by ASSA
c) the computational method is not explained transparently and lacks statistical rigor.
d) no experimental support for the CHASSER using techniques like CHIRP qPCR etc.
I regret to say that despite this tool having potential to be very useful to lncRNA biologists, in the present form is not robust and convincing. Therefore I cannot recommend the manuscript for publication.
Author Response
=== Reviewer 2 ===
The authors in this work use their previously developed tool ASSA to
identify direct RNA-RNA interactions of lncRNAs. While development of
such tool to make accurate predictions is of great interest for the
lncRNA biology field, the paper in the current form under delivers
what is claimed in the abstract.
- a) The writing is not lucid particularly the results and method section.
Response:
We would like to thank the reviewer for drawing our attention to this issue.
We updated the text in the results and methods to make it easier to read. We also added several paragraphs in response to reviewers’ comments. We believe that adding these details make the logic of the paper more lucid.
Also, the original manuscript was sent to a professional English language proofreading specialist. More than 20 minor changes have been corrected. We believe that this will facilitate better understanding of the manuscript.
- b) ASSA should be applied to well-studied lncRNAs like Xist, rox for
which CHIRP datasets are available to validate the targets predicted
by ASSA
Response:
Previously (Antonov et al, 2019, Brief in Bioinf), we already performed the benchmark of ASSA performance in the several well studied lncRNA data sets (RIA-seq for TINCR lncRNA, SPLASH for 50 different RNAs as well as 17 lncRNA-RNA interactions from selected publications). We compared the results obtained by ASSA with 17 different RNA:RNA prediction tools. Although accurate computational prediction remains tricky, ASSA outperformed all other tested tools.
The reviewer suggests using ChIRP-seq to validate the predictions. Yet, this method has been developed to capture RNA:chromatin interactions. Despite that this method can also capture co-transcriptional RNA:nascent RNA interactions, this data would provide only an indirect support for our hypothesis, since ChIRP-seq could not distinguish between various ways of RNA:chromatin interactions. We believe that this analysis is out of the scope of the current work.
- c) the computational method is not explained transparently and lacks
statistical rigor.
Response:
We do understand that ASSA is not a widely used tool in the field at the moment. Still, this computational method is described in detail in the original publication (Antonov et al, 2018, Journal of bioinformatics and computational biology). Moreover, its comprehensive benchmarking can be found in (Antonov et al, 2019, Brief in Bioinf). These two papers are referenced in the current manuscript. Therefore, we believe that description of the algorithm is outside the scope of the current manuscript and would like to direct the interested readers to the corresponding papers for more details.
- d) no experimental support for the CHASSER using techniques like CHIRP qPCR etc.
Resposnse:
We thank the reviewer for the suggestion to use existing experimental methods to check the validity of our bioinformatics predictions. We attempted to do this by analyzing the known interactions of CHASERR from the RISE database (http://rise.zhanglab.net; Gong et al., Nucleic Acids Res 2017; use LINC01578 as a query). Indeed, there were 9 interactions for CHASERR that have been identified experimentally by the PARIS method. These interactions corresponded to 7 unique genes (MEGF8 and NUP153 had 2 different interacting regions). Two of these 7 genes (TNPO2 and NUP153) were present among the 3599 differentially expressed genes (hypergeometric test p-value = 0.0136) identified in the two successful CHASERR knockdowns in the FANTOM6 project. Moreover, NUP153 had two different interacting sites confirmed by PARIS and was also identified as differentially expressed in one of the FANTOM6 knockdowns. It should be noted that ASSA predicts statistically significant interactions between CHASERR and both the TNPO2 and NUP153 transcripts (with p-values 2.45e-08 and 1.66e-12 respectively). Unfortunately, PARIS method was performed in Hela and HEK293T cells, while FANTOM6 pilot data was obtained in fibroblasts. To the best of our knowledge cell-type specificity of RNA:RNA contacts has never been thoroughly investigated. Yet, lncRNA expression is highly tissue specific suggesting that contacts obtained in one cell type may not be preserved in another one. On top of that, not all the contacts may lead to functional consequences such as change of gene expression. Given that, the presence of experimental support for two genes is remarkable.
We added the following text to the paper
to the results:
Finally, it has been experimentally shown that CHASERR interacts with transcripts of 7 different genes in Hela and HEK293T human cells \cite{Lu:2016aa, gong2018rise}. Two of these 7 genes (TNPO2 and NUP153) were present among the 3599 differentially expressed genes identified in the two successful CHASERR knockdowns in the FANTOM6 project (hypergeometric test p-value = 0.0136). It should also be noted that ASSA predicted statistically significant interactions between CHASERR and transcripts of these two genes (with p-values 2.45e-08 and 1.66e-12 respectively).
to the discussion:
Unfortunately, experimental interactions of CHASERR detected by PARIS method were observed in Hela and HEK293T cells, while FANTOM6 pilot data was obtained in fibroblasts. To the best of our knowledge cell-type specificity of RNA:RNA contacts has never been thoroughly investigated. Yet, lncRNA expression is highly tissue specific suggesting that contacts obtained in one cell type may not be preserved in another one. On top of that, not all the contacts may lead to functional consequences such as change of gene expression. Given that, the presence of experimental support for two genes TNPO2 and NUP153 is remarkable.
Finally, the experimental results obtained by an independent study (Siggens et al., Epigenetics Chromatin 2015) suggested that CHD2 was indeed recruited to the promoters of the novel trans targets. However, the details and the targeting mechanism remained unknown. Thus, our bioinformatics predictions shed new light on this process and suggested an important role for the CHASERR lncRNA.
We added the following text to the Discussion:
Finally, it has previously been shown experimentally that CHD2 had distinct binding profiles across active TSS regions. The authors have concluded that transcription-coupled recruitment of CHD2 occurs at transcribed gene TSSs. Our bioinformatics analysis led us to the same conclusion and highlighted the role of \textit{CHASERR} lncRNA in this process.
I regret to say that despite this tool having potential to be very
useful to lncRNA biologists, in the present form is not robust and
convincing. Therefore I cannot recommend the manuscript for
publication.
Response:
We hope that the changes we made in our paper and explanations we provided in response to reviewers comments will change how the reviewer feels about our work.
Round 2
Reviewer 2 Report
Methods section should be divided under proper sub headings.
The github link to the algorithms used and list of previously published datasets list should be included in the methods.
Author Response
Methods section should be divided under proper sub headings.
Response:
DONE
=====
The github link to the algorithms used and list of previously published datasets list should be included in the methods.
Response:
We added a separate section in the method to briefly introduce ASSA and to mention our previous benchmarking work. Particularly, the following text has been added to the Methods:
The AntiSense Search Approach (ASSA, \url{https://github.com/vanya-antonov/assa}) tool was used in this study to predict inter-molecular RNA:RNA interactions \cite{antonov2018assa}. We decided to use ASSA because previously its performance has been evaluated on several well studied lncRNA data sets (RIA-seq for TINCR lncRNA \cite{kretz2013control}, SPLASH for 50 different RNAs \cite{Aw:2016aa} as well as 17 lncRNA:RNA interactions from selected publications). We compared the results obtained by ASSA with 17 different RNA:RNA prediction tools. Although accurate computational prediction remains tricky, ASSA outperformed all other tested tools (see \cite{antonov2019prediction} for more details).
We also corrected several typos in the manuscript text.